# Self-Repairing Herpesvirus Saimiri Deletion Variants

**DOI:** 10.3390/v14071525

**Published:** 2022-07-13

**Authors:** Ines Heyn, Linda Bremer, Philipp Zingler, Helmut Fickenscher

**Affiliations:** Institute for Infection Medicine, Christian-Albrecht University of Kiel, University Medical Center Schleswig-Holstein, Brunswiker Str. 4, 24105 Kiel, Germany; heyn@infmed.uni-kiel.de (I.H.); linda.bremer.2012@gmail.com (L.B.); philippzingler@googlemail.com (P.Z.)

**Keywords:** herpesvirus saimiri, vector, recombination

## Abstract

Herpesvirus saimiri (HVS) is discussed as a possible vector in gene therapy. In order to create a self-repairing HVS vector, the F plasmid vector moiety of the bacterial artificial chromosome (BAC) was transposed via Red recombination into the virus genes *ORF22* or *ORF29b*, both important for virus replication. Repetitive sequences were additionally inserted, allowing the removal of the F-derived sequences from the viral DNA genome upon reconstitution in permissive epithelial cells. Moreover, these self-repair-enabled BACs were used to generate deletion variants of the transforming strain C488 in order to minimalize the virus genome. Using the *en passant* mutagenesis with two subsequent homologous recombination steps, the BAC was seamlessly manipulated. To ensure the replication capacity in permissive monkey cells, replication kinetics for all generated virus variants were documented. HVS variants with increased insert capacity reached the self-repair within two to three passages in permissive epithelial cells. The seamless deletion of *ORFs 3/21*, *12–14*, *16 or 71* did not abolish replication competence. Apoptosis induction did not seem to be altered in human T cells transformed with deletion variants lacking *ORF16* or *ORF71*. These virus variants form an important step towards creating a potential minimal virus vector for gene therapy, for example, in human T cells.

## 1. Introduction

Herpesvirus saimiri (HVS) is the prototype member of the rhadinoviruses, the γ_2_ genus of the family *Herpesviridae.* While its natural host, the squirrel monkey *Saimiri sciureus*, does not show symptoms of infection, other species suffer from lymphoproliferative disorders upon experimental infection [1,2,3,4]. In 1985, as one of the earliest gene therapy experiments, HVS vectors were successfully applied, resulting in the recombinant expression of bovine growth hormone in primates [5]. The transformation capability of HVS in primate T cells was soon discovered to extend to human T cells as well [6,7]. Members of the subgroup C of HVS were shown to be able to transform human T cells without establishing productive infection [8]. For these reasons, HVS is a vector candidate for human T cell gene therapy. The benefits it offers are large insert capacities and episomal persistence, leading to stable transformation of human T cells [9]. The episomal persistence at copy numbers of 50–300 episomes per cell eliminates the risk of unwanted mutations by insertion into the host chromosomes and ensures stable distribution to progenitor cells upon cell proliferation [8,10]. Persistence in proliferating cells was found to be mediated by *ORF73* and the terminal repeat region called H-DNA [11,12,13]. While the non-coding H-DNA consists of repeats of approximately 1.4 kb with a high GC content, the coding L-DNA consisting of 75 identified ORFs has a low GC content [14,15]. Due to varying numbers of the H-DNA repeat units, the genome size can vary. Usually, it is reported to vary from 130–160 kb of which 110 kb constitute the L-DNA [14]. Transgene transcription can be maintained stably over more than four years in culture, depending on the chosen promoter, locus of insertion and the transcript [10]. Therefore, we aimed to generate self-repairing minimal variants with increased insert capacity and replication kinetics similar to the wild-type (WT) virus in order to achieve high-titer virus stocks.

Historically, manipulations of viral genomes became much easier with the establishment of bacterial artificial chromosomes (BACs) in 1997 in which the bacterial F plasmid vector moiety (miniF) serves as a bacterial vector backbone [16]. It was further improved in 2006 when the two-step *en passant* mutagenesis was first described [17]. In the specifically designed *E. coli* strain GS1783, BACs could be introduced and subsequently manipulated based on Red recombination in a seamless and highly efficient manner. Modified versions of this method were used to establish self-repairing varicella zoster virus variants [18,19]. Repetitive viral DNA sequences were introduced into the miniF sequence and enabled homologous recombination upon reconstitution of the virus in permissive cells, causing the vector’s release from the viral genome, leaving it only with the desired alterations. Choosing a viral gene involved in replication should increase the chances of viruses with recombined and self-repaired genomes to replicate faster and gives them an advantage over non-repaired viruses.

For this study, we chose *ORF22*, coding for the viral glycoprotein H, and *ORF29b*, which encodes the HVS terminase [14,20,21]. Glycoprotein H is highly conserved among the Herpesviridae, most likely due to its importance in cell-to-cell spread and infection [20]. gH was reported to form heterodimers with another essential glycoprotein, gL. Together with gB, these glycoproteins are reported to mediate fusion of the virion with the host cell membrane [22]. Disruption of the *ORF22* would, therefore, most likely inhibit viral entry and, thus, viral replication.

Herpesvirus genome packaging is usually performed by a terminase, a complex of several proteins [23]. For HVS, *ORF29* was reported to be part of this packaging complex [21]. The viral gene carries a large intron which is conserved over the mammalian herpesviruses. *ORF29a* indicates the first exon, encoding the N-terminal segment of the protein, while the second exon was designated *ORF29b* and codes for its carboxy-terminus [14]. The terminase cleaves the double-stranded DNA concatemer, which results from the rolling circle replication of the herpesvirus genomes. The multicopy strand is cleaved behind the second terminal repeat during packaging of the DNA into a procapsid, ensuring that only one genome copy is stored within [23,24]. Without the terminase encoded by *ORF29*, cleavage of the multicopy DNA strand would be prevented and packaging impeded, resulting in a loss of infectious virions.

Insertion of the miniF in either ORF results in its disruption as long as the viral genome is cultivated in the form of BACs in bacteria. However, once it enters a permissive eukaryotic cell, homologous recombination should occur, resulting in the complete, seamless restoration of the respective ORF.

## 2. Materials and Methods

### 2.1. Generation of Self-Repairing BACs

All modifications described further on were based on the Red-mediated *en passant* mutagenesis [17,19]. The Red genes of the recombination apparatus of a defective lambda phage were located under control of a temperature-sensitive promoter while the expression of the endonuclease I-SceI was controlled by an arabinose-sensitive promoter [25]. Upon an I-SceI-mediated double-strand break, Red recombinases initiate recombination between homologous regions of 50–80 kb. For all experiments, the BAC of HVS C488 (GenBank entry AJ410493.1) was used in the *E. coli* strain GS1783 [17,21]. The single-copy miniF sequence ensures propagation and maintenance of the viral genome in bacteria while preventing unwanted recombinations due to at most two copies per cell at a time.

Originally, the miniF sequence was located within *ORF14* [26]. To improve efficiency of the miniF release, the sequence was relocated into ORFs important for viral replication. Two different loci were chosen, *ORF29b* (nt. 48,089–49,231; position 530) and *ORF22* (nt. 36,775–38,928; position 1113), to be disrupted and carry the bacterial backbone. Simultaneously, *ORF14* was restored via another homologous recombination [27].

To briefly describe the experimental process, the selection marker, a chloramphenicol resistance cassette (*cat*), was exchanged using Red-mediated recombination and was replaced with a kanamycin (Kan) resistance gene (*aphA1*) from pepKanS2 [17]. The *aphA1* was amplified using the primer 5′-CGGGCG TAT TTT TTG AGT TAT CGA GAT TTT CAG GAG CTA AGG AAG CTA AAC GAT TTA TTC AAC AAA GCC A-3′ and 5′-AACATG TGA GGT TTG ATA GCT TGT CTT ATT GGG AAA GAA TTA AAA GAT CTG GTA CCC CAT GGT GTA TAA AGA CAC CAA GTG AAA AAG CAG CAA TAT TGA ACT AGG GAT AAC AGG GTA ATG CC-3′. The reverse primer generated a duplication segment homologous to *ORF14* upstream of the miniF sequence due to which it could later be released upon I-SceI expression and Red gene induction.

A transfer vector was cloned, carrying the miniF sequence derived from pBeloBamH1 [19] and flanked by 0.5 kb regions homologous to the target sequence of *ORF29b* or *ORF22* on either side. The vector was linearized by restriction enzyme digestion. Bacteria carrying the HVS C488 BAC were made electro- and recombination competent and, synchronously, the modified miniF-Kan sequence was released from *ORF14* while the miniF sequence with *cat* was inserted into *ORF29b* or *ORF22* via the transfer vector. To enable the seamless removal of the miniF sequence from the viral genome upon reconstitution, a second transfer vector was created. This vector additionally carried the inverted (DX) or direct repeat (DR) regions necessary for the self-repair, as well as the *aphA1* gene for positive selection downstream of the *cat* gene within the miniF sequence. This second vector underwent restriction enzyme digestion with I-CeuI, leaving 0.6 kb upstream of DX/DR and downstream of the *aphA1*. The resulting fragment was inserted into the repositioned miniF sequence within the viral genome. Upon virus reconstitution in permissive cells, the inserted repeats could recombine with their direct (*ORF22*) or inverted (*ORF29b*) homologous counterparts up- and downstream of the miniF sequence, leading to the seamless release of the miniF sequence from the viral genome in two steps and to a restored *ORF29b* and *ORF22*, respectively. Both homologous repeats had 0.5–0.7 kb. All virus variants were checked by restriction-fragment length polymorphism analysis (RFLP), polymerase chain reaction (PCR), and Sanger sequencing of the altered ORFs or regions of interest. An extensive scheme is available in the supplement (Appendix A: Scheme of complete procedure for generation of self-repairing HVS BAC variant WT29fDX).

### 2.2. Generation of Self-Repairing Minimal Variant BACs

The deletion variants were generated using the *en passant* mutagenesis as well, replacing the respective ORF with the *aphA1* amplified by PCR from pepKanS2 using the following primers (PAGE purified; Biomers, Ulm, Germany): *ORF3* 5′-GCT GTA TTT AAA AAC TAT TGT TTA ATT ATT AAA GTC AAT TGC TGA ACA AGA CTT TTG TGG CAG AAA CTT ATG TAA CTT TAA GTG CAT TAA TTT AGG GAT AAC AGG GTA ATC GAT TTA-3′ and 5′-GCA CAA GAT ATG CTA TAT AAG TGA ATT TAG TAT AAT TAA TGC ACT TAA AGT TAC ATA AGT TTC TGC CAC AAA AGT CTT GTT CAC AAC CAA TTA ACC AAT TCT GAT TA-3′, *ORF12-14* 5′-TGACTT ATC AAC ATA CAA ATA AAA AAT TTT CAG AAA CAC TAA GTC TAG TTT TAA AGT TAG AAT GAC TTT ACA TTG TAG GGA TAA CAG GGT AAT CGA TTT A-3′ and 5′-ATTTAC ATT GTA AAC TAT ATA TAG GCA ATG TAA AGT CAT TCT AAC TTT AAA ACT AGA CTT AGT GTT TCT GAA AAT CAA CCA ATT AAC CAA TTC TGA TTA-3′, *ORF16* 5′-TGT TTA ACA AGC ATA TTC ATA ACA GCA GCT GAG TTA CCA CAT CTA AGA AAA ACT CCT CAA CTA TTA TTA ATT TTT TAG GGA TAA CAG GGT AAT CGA TTT-3′ and 5′-GAT GAA TTG TTG AAT AAA TAG ACA TAA AAA TTA ATA ATA GTT GAG GAG TTT TTC TTA GAT GTG GTA ACT CAG CTG GAA AAA CTC ATC GAG CAT CA-3′, central *ORF21* 5′-TTCCTA TAA CTG CTT CTA ATA GTG TGT CCG AGC TTT TAA GTC TAC ATG ATC CTG AAG AAA TTG TAG AAG TAT GTT TAG GGA TAA CAG GGT AAT CGA TTT A-3′ and 5′-AGATCT GTT ATG TGC TTT GCA TTG AAA CAT ACT TCT ACA ATT TCT TCA GGA TCA TGT AGA CTT AAA AGC TCG GAC CAA CCA ATT AAC CAA TTC TGA TTA-3′, *ORF71* 5′-AGTACA GAC ATT TCA AAT AAC TTA TTA TAA ACC ACA TTC ATG CTA GAC TAT TAC AGA ATT TCG AGG TCA TAT AAA TAG GGA TAA CAG GGT AAT CGA TTT-3′ and 5′-AAGATC AAG TGT CTG AAG CAT TTT CTT TAT ATG ACC TCG AAA TTC TGT AAT AGT CTA GCA TGA ATG TGG TTT ATA GAA AAA CTC ATC GAG CAT CA-3′.

### 2.3. Cell Culture, Replication Kinetics and Titration

Permissive epithelial owl monkey kidney (OMK) cells (American Type Culture Collection, Manassas, VA, USA; CRL1556) were cultured in Dulbecco’s modified Eagle’s medium (DMEM) with phenol red; with 3.7 g/L NaHCO_3_, 4.5 g/L D-glucose (Bio&Sell, Feucht, Germany), 10% fetal bovine serum, 1% L-glutamine and 1% penicillin/streptomycin. Cells were split 1:2 once a week and medium exchanged every 3–4 d.

For transfection, 2 × 10^5^ OMK cells were seeded in three respective wells of a 6-well plate and transfected using Lipofectamine2000 (Fisher Scientific, Schwerte, Germany) with BAC DNA isolated via a Maxi preparation kit (Qiagen, Hilden, Germany). Cells from all three wells were pooled and transferred 3–6 d later into a 75 cm^2^ flask and grown until the viral lysis was completed.

The replication kinetics were initiated by seeding 4 × 10^4^ OMK cells per well of a 24-well plate and infecting them the following day with a multiplicity of infection of 0.01 for 1 h with the respective virus. Supernatants were collected and frozen at −80 °C, including the inoculum on d 0, every second day from d 1 on until complete lysis was reached (usually d 9, max. d 13). Once harvest was completed, the supernatants were thawed and diluted appropriately to be titrated in 24-well plates with 4 × 10^4^ OMK cells per well. Counting of cytopathic effects was carried out 3–5 d after infection. All viruses were subjected to replication kinetics at least three times, except for WT29fDX∆12-14 which was performed only once.

Viruses WT29fDX, WT22fDR, WT29fDX∆3∆M21, WT29fDX∆12-14, WT29fDX∆71 and WT22fDR∆71 were passaged by infecting a 175 cm^2^ flask with OMK cells (approximately 12 × 10^6^ cells) with 500 µL of a virus stock with titers of 10^5^–10^6^ plaque-forming units (pfu) per ml in 6 mL. Infection lasted for 1 h at 37 °C, 95% humidity and 5% CO_2_ and the flask was moved regularly to prevent the cells from drying out. Subsequently, the medium was removed completely and fresh medium was added. The flask was incubated until lysis was complete and all cells were dead, and the supernatant was collected, centrifuged to remove cellular debris and stored at 4 °C until further use. Virus WT29fDX∆16 was passaged by adding 500 µL of a virus-containing supernatant with titers of 10^5^–10^6^ pfu/mL to OMK cells seeded in a 75 cm^2^ flask in 12 mL medium. The flask was incubated until full lysis was reached and virus supernatant collected similarly as described above.

Viral DNA was isolated from supernatants of OMK cells by centrifugation of 6 mL for 2 h at 21,000× *g* at 4 °C. The resulting pellet was dissolved in DNaseI mix (Roche, Basel, Switherlands) and digested for 30 min at 37 °C. Inactivation at 75 °C for 15 min and cooling down on ice followed, until the samples reached approximately 25 °C. A volume of 100 µL sterile phosphate buffered saline (Bio&Sell) was added, followed by 20 µL of proteinase K (Qiagen) and 20 µL RNase A (ThermoFisher Scientific, Waltham, MA, USA) and vortexing. The DNeasy^®^Blood & Tissue Kit (Qiagen) was used according to the manufacturer’s instructions. After elution from the column, 20 µL of 3M sodium acetate were added and the samples inverted, then treated with 400 µL 100% *v*/*v* ethanol, inverted again and incubated at −80 °C for 30 min. The samples were subsequently centrifuged at 4 °C for 1 h at 21,000× *g*, the supernatant removed and the DNA resuspended after air-drying in 17 µL nuclease- and DNA-free water (not DEPC treated, Genaxxon, Ulm, Germany).

The reference genome of HVS C488 referred to in this paper can be found under GenBank entry AJ410493.1.

### 2.4. PCR Monitoring of the Self-Repair

Virions were isolated from supernatants by centrifugation for at least 2 h at 4 °C and 21,100× *g*. The cell culture medium was removed and virions resuspended in RNase- and DNase-free water. The HotStarTaq^®^ DNA polymerase (Qiagen) was used to amplify regions of interest in order to prove self-repair and confirm consistency of the established deletion of selected ORFs: *ORF75* 5′-ATGCTT TTA GTA GTT TGA GG-3′ and 5′-AGGCAC AGT TGA CAA TGT-3′, *sopA* 5′-TGGGGT TTC TTC TCA GGC TAT C-3′ and 5′-TAGTCA AAC AAC TCA GCA GGC G-3′, *cat* 5′-TGCCAC TCA TCG CAG TAC TG-3′ and 5′-AGGCAT TTC AGT CAG TTG CTC-3′, *ORF3* 5′-TGCCTA AGT ACC TAG TGC C-3′ and 5′-CCCTTT ATA CAA GAC TAA AA-3′, *ORF12-14* 5′-TCTGCT ATC TGT TTG CCT G-3′ and 5′-GAAACT GAC ACA TAT TAT GAG CC-3′, *ORF16* 5′-GCCAAA CTT GCC AGT TAA TTA CAT-3′ and 5′-CCTACT GAA AGT GCA TCC AAT GAA-3′, central *ORF21* (*M21*) 5′-ATGACC GGA AGA GGA CAG CCT C-3′ and 5′-TCATTG AGA ATT AAA CGT CCT CGC-3′, *ORF22* 5′-GTAAGG GCA AGC GTC ACT C-3′ and 5′-ATGAAT TAG TTG AGC TAT GGC-3′, *ORF29b* 5′-AACGAG TAG GAG GAA AGT C-3′ and 5′-TTGAAA ATG TGT TTG TTT GAG G-3′, *ORF71* 5′-GCTTCA TTG CTT TTC TAC TAC TCT-3′ and 5′-TTGGAC TTG TTA CCT AGA AGA CC-3′.

### 2.5. Transduction and Cultivation of Native and Transformed Human T Cells

Human T cells from buffy coats from whole blood (stored at −80 °C in fetal bovine serum (FBS) with 9% *v*/*v* dimethyl sulfoxide (DMSO)) were thawed and diluted with RPMI 1640 medium (Bio&Sell), centrifuged at 260× *g* for 10 min and resuspended in RPMI 1640 (10% FBS, 1% L-glutamine, 1% penicillin/streptomycin). The cells were activated using concanavalin A at 4 µg per ml medium. The following day, the medium volume was matched with the same volume of TC50 medium (44% Panserin^®^401 (PAN, Aidenbach, Germany), 44% RPMI 1640; 10% FBS; 1% penicillin/streptomycin; 1% L-glutamine; 50 U/mL interleukin 2 (Clinigen Healthcare B.V., Schiphol, The Netherlands)). Three days later, activated T cells were transduced at a multiplicity of infection (MOI) of 3 via spinoculation (460× *g*, 26 °C, 2 h). Subsequent culturing included partial medium changes (approx. 30–40%) every 3–4 days in TC50 and maintenance of stable cell density of 1–2 × 10^6^ cells/mL.

### 2.6. Apoptosis Induction in Transformed Human T Cells and Flow Cytometry Data

First, 1 × 10^6^ viable cells were harvested, counted and washed with PBS at room temperature (50× *g*, 10 min). Apoptosis was initiated in phenol red-free RPMI 1640 (10% FBS, 1% L-glutamine, 1% penicillin/streptomycin, Bio&Sell) using hydrogen peroxide (final concentration 200 µM) or topoisomerase inhibitor camptothecin (final concentration 6 µM; Merck, Darmstadt, Germany). The six different cell lines (K38 or K39, transformed with either WT(29fDX), WT(29fDX)∆16 or WT(29fDX)∆71) and non-transformed donor T cells were incubated for 4 or 24 h (37 °C, 5% CO_2_, relative humidity 95%) and subsequently washed in PBS (260× *g*, 10 min). FITC Annexin V Apoptosis Detection Kit I (Becton Dickinson, Heidelberg, Germany) was used for the staining of apoptosis markers. Compensation was performed using single stained FITC and PI controls of transformed K39 WT(29fDX) T cells at a low flow rate.

Flow speed was set to low and raw data for forward scatter, side scatter, FITC-A and peridinin chlorophyll protein-cyanine 5.5 (PerCP Cy 5.5) were generated. The FACS^®^ Canto I instrument was used with its laser Sapphire^®^ Solid State 488 20, preventing doublet exclusion due to technical limitations. Emission of FITC was measured using the 530/30 bandpass filter, PI-emission was measured using the 670 long-pass mirror. For all samples, the area under the curve was measured, indicated by A. During flow cytometry, the gate was set to stop when data of 5 × 10^4^ living cells were collected. Data evaluation and plots were generated using FCS express 6 (De Novo Software, Pasadena, CA, USA).

### 2.7. ORF9-Specific Realtime PCR and HVS Genome Copy-Number Quantitation

Total DNA of 1 × 10^6^ transformed T cells was isolated (DNeasy Blood & Tissue kit, Qiagen) and realtime PCR performed on a 7500 realtime PCR system (ThermoFisher Scientific, Waltham, MA, USA) using Fast Advanced Taqman Kit (Applied Biosystems ThermoFisher Scientific, Waltham, MA, USA). HVS *ORF9*-specific PCR was performed using the probe 6fam-TTA CAG GAG TTG CGT CAG GCT TGCT-BMN-Q535, and *ORF9-for* 5′-TTC TAG ATA AAC AGC AAC TCG C-3′ and *ORF9*-rev 5′-TTC TTG AAG TGT ATC AGG TGT C-3′. Human *GAPDH* DNA was detected using *GAPDH*-for 5′-CCC CAC ACA CAT GCA CTT ACC-3′ and *GAPDH*-rev 5′-CCT AGT CCC AGG GCT TTG ATT-3′ as well as *GAPDH*-probe FAM-TAG GAA GGA CAG GCA AC-TQ2 (all oligonucleotides: Biomers). DNA standards established by our group were included. DNA copy numbers were estimated afterwards by verifying cell numbers by use of human *GAPDH* and estimating the viral copy numbers following the suggestions given by the University of Rhode Island of Genomics & Sequencing Center. Assuming an average weight of a base pair (bp) of 650 Da, copy numbers were determined by dividing the product of mass (ng) and Avogadro’s constant (6.022 × 10^23^) by the product of the size (bp) multiplied by 650 and 10^9^, in order to convert g to ng [28].

## 3. Results

### 3.1. Self-Repair of Wild-Type Virus Variants WT29fDX and WT22fDR

The self-repair of the two wild-type (WT) HVS C488 variants depended on the occurrence of two separate homologous recombination events during virus propagation in permissive OMK cells (Figure 1).

The progress of the seamless and complete removal of the miniF viral self-repair was monitored by conventional PCR after each passage in OMK cells (Figure 2a). The primer-binding sites within the viral genome are indicated in Figure 1. The *cat* gene was chosen to monitor the successful recombination upstream of the repeat DR or DX, and *sopA* was chosen to monitor the downstream loss of the second part of the miniF sequence. Self-restoration of *ORF22* and *ORF29b* was also monitored. Both virus variants were found negative for the two miniF fragments no later than passage 2. Accordingly, restoration of the *ORF22* and *ORF29* was increasingly detected as can be seen in Figure 2a. Each viral passage was completed within 7 to 9 d.

Replication kinetics of the two virus variants were documented in OMK cells when passage 9 was complete and paralleled that of the wild-type virus in at least three separate experiments. The data of one exemplary kinetic experiment is displayed in Figure 2b. Usually, it took 7 to 9 d to reach complete lysis for HVS C488 in permissive cells and titers varied from 10^5^ to 10^6^ pfu/mL.

Genomic integrity was confirmed using comparative restriction-fragment length polymorphism analysis (RFLP) with the restriction enzymes EcoRV and NcoI. The restriction enzyme-digested viral BAC genomes, isolated from *E. coli*, were compared to self-repaired viral DNA, isolated from the supernatant of a completely lysed OMK cell culture originally infected with viruses from passage 9 (Figure 3). As expected, the changes between the wild-type virus variant WTf and the self-repairing variants WT22fDR and WT29fDX were clearly visible by band shifts (Figure 3a), while the patterns between self-repaired WT22fDR and WT29fDX did not vary. The coinciding band patterns of self-repaired WT(22fDR) and WT(29fDX) clearly indicate the successful self-repair. Comparison with the predicted fragment pattern using Vector NTI (Invitrogen) revealed that all bands appeared to have the expected size according to agarose gel electrophoresis, suggesting genomic integrity is intact apart from the intended alterations. However, the patterns between WT viral DNA and the self-repaired viruses differed slightly. In particular, the bands at approximately 10 kb in the EcoRV digest and at approximately 6.2 kb in the NcoI digest were prominent differences. Possible reasons for these bands include methylated CpG islands which prevent the cleavage of the viral genome, leaving bigger bands than predicted. Additionally, virus cultures always include polymorphisms which can interfere with predictions and cause alterations from the expected band patterns.

To ensure the genomes did not differ from the expected sequence, the virus WT and the self-repaired virus WT(29fDX) were subjected to next-generation sequencing on an Illumina Novaseq 6000 (Eurofins Genomics, Ebersberg, Germany) and compared to the published sequence after completing passage nine [21]. Sequencing revealed several sequence alterations shared by the two virus variants but also separately occurring point mutations (Table 1). Furthermore, it confirmed that some variations were carried only by part of the virus population, indicated by the exchange frequency. Most importantly, however, sequencing confirmed the correct genome sequence and seamless self-restoration of *ORF29b*. Only the coding L-DNA of the virus genomes was sequenced, since the non-coding H-DNA is characterized by a very high GC content.

Next-generation sequencing confirmed the correct and seamless self-repair of WT29fDX to WT(29fDX) after passage 9. The identified alterations in coding regions are listed in Table 1. Additionally, three further alterations in non-coding regions were observed in the self-repaired virus (nucleotide 4 T > C or single-base deletion, nucleotide 28,380 C > T and nucleotide 29,761 C > T).

### 3.2. Self-Repairing Deletion and Reporter Variants

Based on the confirmed self-repairing virus variants, variants with reduced genome size were generated. *En passant* mutagenesis was used to seamlessly replace the whole ORF by the positive selection marker *aphA1*. In the case of *ORF21*, only the central sequence was replaced (M21) so as to not compromise the promoters for the neighboring ORFs. In a second step, the selection marker was removed from the genome. Most of the targeted ORFs were already established to be dispensable for viral replication and transformation but have not necessarily been seamlessly and completely deleted from the viral genome [29,30,31,32,33]. *ORF3* and the central sequence of *ORF21*, *ORF12-14*, *ORF16* and *ORF71* could be successfully removed from the HVS genome as demonstrated by RFLP (Appendix A: RFLP of indicated manipulated virus variants based on self-repairing BACs WT29fDX and WT22fDR, compared to the wild-type BAC WTf) [10,34]. Reconstitution of the single deletion variant WT29fDX∆3 was not yet possible, although recovery of a virus variant derived from it was possible, with both *ORF3* and the central part of *ORF21* deleted. The reporter vector WT29fDXHTLV carried an *mRFP*-reporter gene under control of the HTLV promoter between the *ORF75* promoter and H-DNA at the right-most end of the L-DNA.

PCR confirmed the self-repair and restoration of either *ORF22* or *ORF29b* in passage 3 at the earliest as can be seen for WT29fDX∆3∆M21, and at the latest during p7 as demonstrated by WT29fDX∆12-14 (Figure 4a). WT29fDX∆16 showed similarly slow restoration of *ORF29b* upon passage 7 but could not be compared directly to the other deletion virus variants since it was processed differently (Appendix A: PCR monitoring of the self-repair of WT29fDX and WT29fDXΔ16). Comparison of the two deletion variants of *ORF71* in either WT22fDR or WT29fDX revealed that self-repair was achieved faster in WT29fDX∆71.

All vector variants were shown to have similar replication kinetics to the wild-type (Figure 4b). The replication of WT(29fDX)∆16 was found to match that of the wild-type as well [10]. The double deletion variant WT(29fDX)∆3∆M21 singularly replicated slower and completed lysis at lower titers compared to the wild-type (Figure 4b).

Double deletion of *ORF21* combined with differing ORFs in a non-repairing vector did not result in the same reduction in end titer and deceleration in replication but single deletion of either *ORF3* or *ORF75*, both encoding for viral formyl-glycinamide phosphoribosyl-amidotransferase (FGARAT), decreased replication efficiency similarly to what was observed for WT(29fDX)∆3∆M21 and in accordance with previous studies [10,26,29]. The replication kinetics shown in Figure 4 resulted from one experiment; normalized kinetics of three individual experiments can be found in the supplement (Appendix A: Exemplary normalized replication kinetics of generated virus variants, Appendix A: Examples of replication kinetics of different virus variants at different MOIs between 0.01 and 0.5).

Vector persistence in transformed human T cells was found to be stable for at least 32 months after transduction. T cells derived from two different donors were transduced with an MOI of 3. Cell lines derived from donor K39 showed average copy numbers from 94 up to 130 copies per cell 31 months after initial transduction and continued cultivation. Semi-quantitative realtime PCR, specific for HVS *ORF9*-DNA, was performed in relation to *GAPDH* DNA. Lytic replication in these T cells was tested for by transferring supernatant to permissive OMK cells and signs of virus infection could not be observed.

As a side note, we report that virus variants with complete and precise deletion of the single *ORFs 45*, *49*, *52*, *62*, *63*, *64*, *69* or *70* from the wild-type genome could not be reconstituted in at least three separate experiments (Appendix A: RFLP of indicated manipulated virus variants compared to the wild-type BAC WTf) [10]. These single deletions were found to result in abortive infection upon transfection into permissive OMK cells.

### 3.3. Average Copy Numbers per Cell and Apoptosis Induction in Human T Cells Transformed with Self-Repairing HVS Deletion Variants

Human T cells from two different donors (K38, K39) were transduced with self-repaired virus variants WT(29fDX), WT(29fDX)∆16 and WT(29fDX)∆71. The transformed cell lines were continuously cultivated over 32 months. We were interested whether the deletion of *ORF16* or *ORF71* would result in changes in the response of the transformed T cells to apoptosis inducers.

An *ORF9*-specific realtime PCR was established to determine the average copy numbers per cell to ensure that HVS persisted at similar copy numbers. *ORF9* codes for the viral polymerase and is deemed essential for viral replication. Human glyceraldehyde-3-phosphate dehydrogenase (GAPDH) was used to quantitate HVS copy numbers in relation to copy numbers of the human T cells. The cell line K39 WT(29fDX) carried 131 copies per cell on average, K39 WT(29fDX)∆16 had 96 copies and K39 WT(29fDX)∆71 94 copies (compare Appendix A: Average HVS-DNA copy numbers in 31-month-old, transformed human T cells of donor K39). All cell lines tested had copy numbers exceeding the initially used MOI of 3 for transduction 31 months prior by far, indicating genomic replication. Meanwhile, release of infectious virions into the supernatant was not detected when testing it on permissive OMK cells.

The cell lines were subsequently used in FITC annexin V/propidium iodide staining and percentages of apoptotic cells measured 4 h and 24 h after apoptosis induction. We chose oxidative stress mediated by hydrogen peroxide (H_2_O_2_) to induce the intrinsic pathway, and camptothecin (CPT) to induce response to DNA damage. In non-transformed, untreated T cells of the same donor lines, percentages of FITC-positive cells were initially high (50–70%) after 4 h and 24 h. Treatment with either inductor increased percentages of FITC-positive cells after 4 h of treatment slightly. After 24 h, percentages reached almost 100% (Figure 5a).

By contrast, transformed T cells of both donors, originally transduced with either self-repairing virus variant, did not reach similar rates of FITC-positive cells upon treatment with either inductor. The number of FITC annexin V-positive cells stayed below 15% at both time points, in both donor lines transformed with either virus variant upon treatment with either inductor. Only minor changes were observable when comparing percentages of one cell line treated with different inductors (Figure 5a and Appendix A: Apoptosis induction in 31-month-old HVS-transformed human T cells after 4 h and 24 h). Mostly, the percentages stayed the same or only slightly increased with treatment. In general, the cells transformed with either virus variant were equally well protected from apoptosis induction by reactive oxygen species and from DNA damage caused by topoisomerase inhibition.

Effects of the single deletions on apoptosis induction in transformed human T cells seemed to be minor, too (Figure 5b). In K38-based cell lines, a deletion of *ORF16* seemed to cause a slight increase in FITC-positive cells of about 5%. In K39-based transformed cells, however, the exact opposite effect was observed in two separate experiments, where a minor decrease of at most 3% was measured. Both donor-based cell lines, transformed with self-repaired deletion variant WT(29fDX)∆71, showed slightly increased percentages of apoptotic cells when compared to the wild-type WT(29fDX) transformed cells.

Taken together, these preliminary data suggest only a minor to no effect mediated on apoptosis induction by oxidative stress or topoisomerase inhibition in human T cells transformed with deletion viruses missing either *ORF16* or *ORF71* when compared to wild-type transformed T cells.

## 4. Discussion

Herpesvirus saimiri has been discussed as a potential vector for gene therapy and anti-tumor therapy [5,30,35,36]. Advantages of HVS as a viral vector are high insert capacity, non-integration, episomal persistence within the cell nucleus at high copy numbers, as well as high titers combined with simple propagation in cell culture. Furthermore, the capability of HVS C488 to transform human T cells could be useful for developing therapeutic approaches for genetic disorders such as severe combined immunodeficiencies.

Research into HVS as a vector became particularly interesting when autologous reinfusion of HVS-transformed macaque T cells was reported to be tolerated well, and did not result in lymphomas or other pathologies usually observed when these monkeys were experimentally infected with HVS [37]. Spread of infectious virus from transformed macaque T cells was observed only exceptionally in these monkeys. The fact that macaques which received transformed T cells were protected against a challenging infection with the same HVS strain, and did not develop lymphomas as the control animals did, was highly promising. An improved HVS C488 vector could therefore be a valuable tool for individualized gene therapy and used to introduce transgenes into patients’ T cells. For example, patients suffering from severe combined immunodeficiencies, the collective term for gene defects concerning immune cell development or function, might benefit from the development of HVS into a gene therapeutic vector. Patients with such defects often die at a young age due to opportunistic infections. The gene therapy currently available for treating these defects relies on retroviral vectors for correcting the genomic aberration. However, insertional mutagenesis still poses a risk when using inserting vectors [38,39,40,41]. Murine leukemia virus vectors have a preference for inserting into transcription start regions or introns, potentially deregulating the new host’s gene expression up to the point of cancer development [42]. Although HVS-based vectors would require a long-term monitoring, similarly to integrating vectors, at least the risk of silencing tumor suppressor genes or accidentally activating proto-oncogenes is expected to be eliminated by their episomal persistence.

The genomic integrity of the vector is a crucial aspect when considering self-repairing viruses for in vivo treatment. The acquired data suggest a stable propagation in dividing cells. Furthermore, sequence alterations have been identified which can be considered single-nucleotide polymorphisms, without consequence for the expressed protein. Most alterations are either silent or lead to an exchange with a similar amino acid (e.g., alanine to valine). The exchange of glycine and glutamate is located within a glycine/glutamate-rich sequence of approximately 140 amino acids [12]. Within this stretch of *ORF73*, both amino acids alternate exclusively in strain C488. The predicted nuclear localization sequences are located at the amino-terminal and the carboxy-terminal region of the gene. Lastly, the sequence alteration within *ORF46* might be of interest due to the importance of the viral glycosylase during ƴ-herpesvirus replication [43]. The detected alteration leads to an amino acid exchange of aspartic acid with asparagine, the first being acidic and presenting negatively charged side chains and the latter exhibiting neutral and polar properties. Thus, an influence on the protein structure cannot be excluded. However, viral replication was clearly functional for viruses carrying this alteration, eliminating a crucial effect of this exchange. Additionally, the catalytical and highly conserved domains have been reported to be located in the carboxy-terminal region of *ORF46* [43]. The described alterations within the wild-type virus and the generated self-repairing virus variant are, therefore, likely to affect neither replication nor protein function.

Increased insert capacity and at the same time an improved safety profile could result from the deletion of non-essential virus genes. Ideally, these conditions should be met without impairing the viral replication. We report two viral vector variants enabled to remove the miniF sequence which is absolutely essential for propagation and manipulation in bacteria, yet dispensable once the virus re-enters its life cycle within permissive cells. Both generated variants WT29fDX and WT22fDR completed self-repair early after reconstitution within two passages on OMK cells. The self-repair alone increased the vectors insert capacity by 8 kb. Furthermore, the deletion of several ORFs did not or only slightly impaired viral replication.

A transgene capacity of approximately 3.8 kb was generated by deleting *ORF12–14* which also neither caused a drop in titer nor impairment in replication time. By studying a spontaneous deletion mutant of strain A11, it was already established that all three ORFs, and possibly also *ORF11*, were lacking transforming properties and were not needed for efficient replication, and were therefore chosen for this study [31,32]. We decided to repeat the deletion in a precise manner afflicting only the intended ORFs and were able to reproduce those findings for a self-repairing vector variant. Despite unchanged replication time or titers, the self-repair of this deletion variant lasted longer than for the other variants and was only completed after passage 7.

The deletion of a central sequence of *ORF21* of approximately 0.9 kb was tolerated well in terms of replication competence. *ORF21* encodes for the viral thymidine kinase and the deletion contained five conserved and putatively functional domains, including the nucleotide binding site [44,45]. Disabling the viral thymidine kinase is mostly associated with a reduced pathogenicity which is the reason why the deletion was included in the experiments [46,47,48,49].

Anti-apoptotic properties were previously reported for *ORF16*, encoding for vBcl-2, and *ORF71*, encoding for vFLIP [9]. The anti-apoptotic properties of *ORF16* were first observed using Vero cells and a Sindbis virus which was encoding the viral Bcl-2 whilst at the same time inducing programmed cell death [50]. Disruption of *ORF16* did not diminish viral replication [33,50]. Similarly, replication was unaltered when *ORF71*-deficient virus was grown in OMK cells [30]. In Table 2, an overview of selected ORFs is provided, which were studied by deletion or disruption, and their importance for different viral features, including viral replication, transforming properties and persistence.

We established self-repairing single- and double-deletion virus variants and found that only deletion of either *ORF16* or *ORF71* is tolerated in the context of replication competence, while deletion of both prevented the recovery of infectious virions [34]. Previous reports that replication is unchanged when either ORF is deleted from the viral genome could be confirmed with self-repairing virus variants (Figure 4b and Appendix A: Exemplary normalized replication kinetics of generated virus variants) [30,33].

Copy numbers in 31-month-old transformed human T cells of two different donors were demonstrated to be comparable to HVS genomes per cell, varying around 100 copies (Appendix A: Average HVS DNA copy numbers in 31-month-old, transformed human T cells of donor K39). Furthermore, preliminary experiments with human T cells transformed with deletion variants WT(29fDX)∆16 or WT(29fDX)∆71 were not found to differ considerably in their response to apoptosis induction compared to cells transformed with the wild-type WT(29fDX) but differed greatly from the response of non-transformed, untreated donor T cells in accordance with a previous study (Figure 5) [10,56]. Stimulation with hydrogen peroxide or camptothecin, and subsequent FITC annexin V/PI staining combined with flow cytometry showed almost 100% apoptotic cells in the non-transformed donor cells after 24 h of treatment, while T cells of the same donor transformed with either of the three virus variants had rates of apoptotic cells decreased to below 15%. These findings affirm the notion, based on experiments with transformed New World monkey T cells, that the effect of vFLIP might be too small to be detected and, moreover, suggests the same for vBcl-2 [30]. In view of the unchanged pathogenicity reported for an *ORF71* deletion virus which was used to infect cottontop tamarins (*S. oedipus*) and compared to infection with the wild-type virus, it is unlikely that by deletion of either *ORF16* or *ORF71* the safety profile of an HVS T cell vector would be improved considerably [30]. However, insert capacity is increased by deletion of either ORF by approximately 0.5 kb.

In the future, the combined deletion of the above-mentioned genes in one single vector might provide an optimized minimal vector. From our experiments and the literature, up to 50 kb of insert capacity within a minimal variant HVS vector seem feasible. With regard to the notion that gene silencing in a viral vector is best avoided by using the original cellular promoter of the specific transgene, such an increase in insert capacity might be an important contribution to the vector development [57].

## Figures and Tables

**Figure 1 viruses-14-01525-f001:**
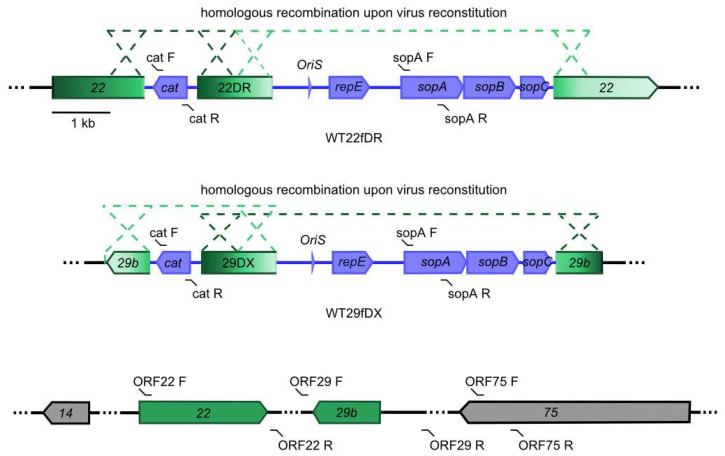
Scheme of the two separate recombination events which constitute the viral self-repair of herpesvirus saimiri (HVS) virus variants within permissive epithelial cells. In the upper panel, WT22fDR carries direct repeats of *ORF22* while in the middle panel, WT29fDX carries inverted repeats of *ORF29b* (indicated by color gradients). The primer combinations used to monitor viral self-repair by PCR are indicated (F: forward primer, R: reverse primer). In the lowest panel, the self-repaired virus genome is shown without F plasmid vector moiety (miniF) sequence (orange-colored genes). The resulting DNA is free of residual base pairs and the ORFs previously harboring the miniF are restored to their original state. Images taken and modified from [27].

**Figure 2 viruses-14-01525-f002:**
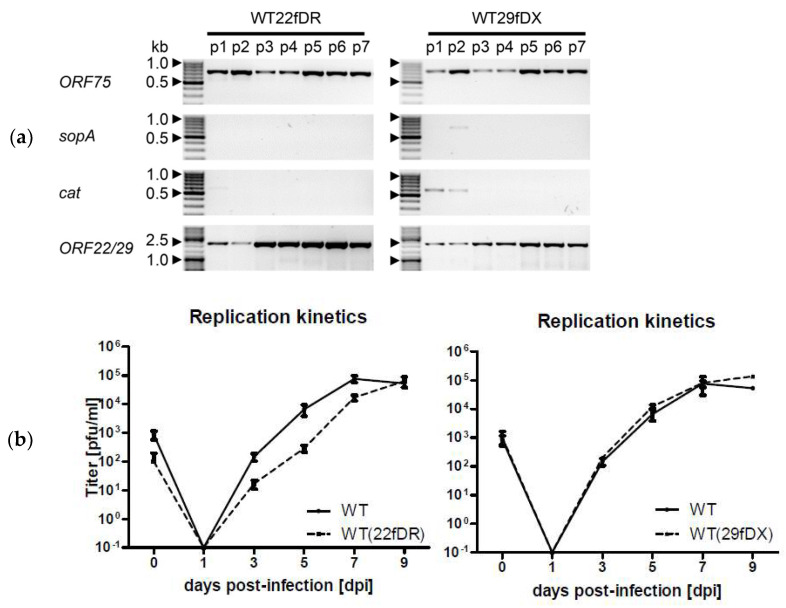
Genomic monitoring and replication properties of the self-repairing virus variants WT22fDR and WT29fDX. (**a**) PCR analysis of viral genomes during passaging of the virus variants in permissive OMK cells. *ORF75* served as a viral control, *sopA* and *cat* were parameters for the respective part of the miniF that was to be lost, and *ORF22/29* PCR indicated self-restoration of the ORF harboring the miniF sequence. Inverse gray shades are depicted. All band sizes were of the expected size. (**b**) Exemplary replication kinetics of one experiment of at least three separate experiments of the self-repaired virus variants after passage 9 (p9). The harvested supernatants were titrated in three individual dilution series of which mean and standard deviation are indicated. Virus titers are given in plaque-forming units (pfu)/mL in relation to the day they were harvested.

**Figure 3 viruses-14-01525-f003:**
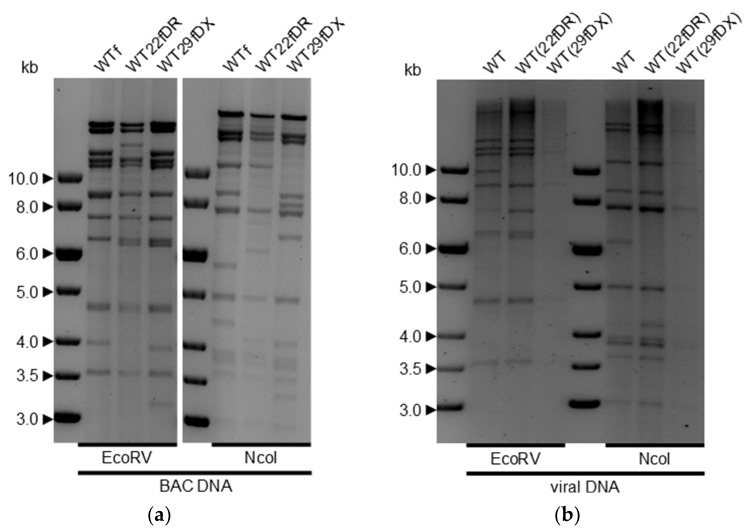
Restriction-fragment length polymorphism analysis (RFLP) of isolated HVS C488 variant DNA subjected to restriction digestion with enzymes EcoRV and NcoI, respectively. The DNA fragments were separated on a 0.7% agarose gel. Gray shades are displayed inversely. (**a**) BAC DNA of the varying wild-type-like virus variants with miniF located in *ORF14* (WTf), *ORF22* (WT22fDR) and *ORF29* (WT29fDX), respectively. (**b**) Viral DNA isolated after nine passages in OMK cells compared to the wild-type virus HVS C488. Self-repair was reliably completed at that passage and the miniF sequence removed from the viral genome.

**Figure 4 viruses-14-01525-f004:**
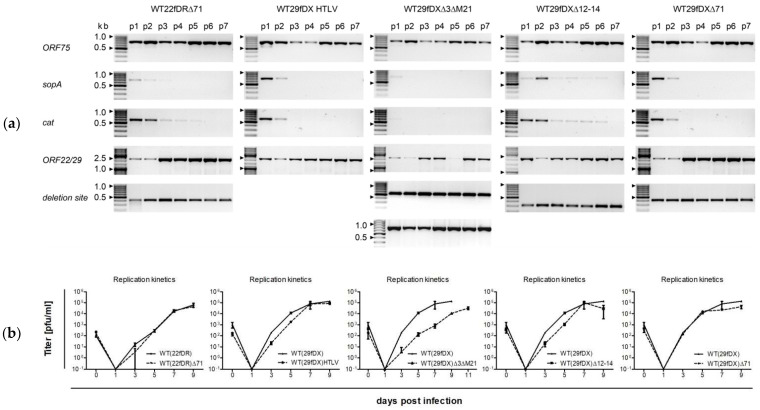
Self-repair of deletion variants and replication kinetics in self-repaired state. (**a**) Conventional PCR after virus passage (p) 1 to 7. *ORF75* PCR served as viral control, *sopA* and *cat* as indicators for the self-repair, as well as the PCR for *ORF22* and *ORF29*, respectively. For the variant WT29fDX∆3∆M21, deletions of *ORF3* and *M21* were proven. WT29fDXHTLV was checked by sequencing instead after passage 9 and was found to be correct. All bands were of the expected size, indicated in kilobase pairs (kb). Gray shades are depicted in inverted mode. (**b**) Replication kinetics of the different deletion variants are plotted compared to the wild-type virus or the self-repaired wild-type parental virus variant. Mean titers and standard deviation are indicated.

**Figure 5 viruses-14-01525-f005:**
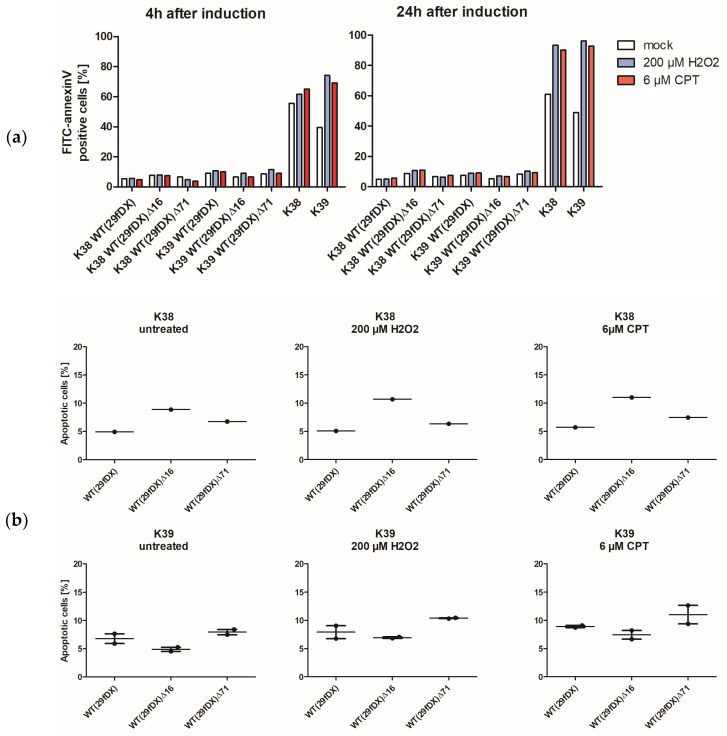
Scatter-plots of FITC-positive cells in percent indicating annexin V-presenting cells. (**a**) Percent of apoptotic cells in transformed cells of donors K38 and K39. Self-repaired wild-type virus WT(29fDX) or a deletion variant with either *ORF16* or *ORF71* deleted were initially used to transform the cells 31 months prior. Effects of apoptosis inducers hydrogen peroxide (H_2_O_2_) or camptothecin at indicated final concentrations were investigated 4 h and 24 h post-induction. (**b**) Effects of the two inductors compared between cells of the same donor, transformed with the different self-repaired virus variants 24 h post-induction. Cell numbers did not suffice for more than one (K38 cell lines) and two experiments (K39 cell lines), respectively.

**Table 1 viruses-14-01525-t001:** Sequence alterations found by next-generation sequencing in coding regions of isolated virus genomes.

WT			Self-Repaired WT(29fDX), p10		
Site of Alteration	ORF	Exchange Frequency (% of Reads)	Site of Alteration	ORF	Exchange Frequency (% of Reads)
			46,379 G > A Leu216Leu	ORF26, capsid protein VP23	32
55,412 T > G Leu402Leu	ORF36, protein kinase	30			
66,446 C > T Asp65Asn	ORF46, DNA glycosylase	100	66,446 C > T Asp65Asn	ORF46, DNA glycosylase	100
66,884 G > A Ala50Val	ORF47, glycoprotein L	50	66,884 G > A Ala50Val	ORF47, glycoprotein L	50
106,869 C > T Gly149Glu	ORF73, LANA homolog	22	106,869 C > T Gly149Glu	ORF73, LANA homolog	36
			106,872 T > C Glu148Gly	ORF73, LANA homolog	30

**Table 2 viruses-14-01525-t002:** Selected HVS C488 ORFs.

*ORF*	Function/Protein [21]	Essential for	Size (kb)	Source
*1*	stpC/Tip	transformation	1.4	[51,52]
*3*	vFGARAT, induction of proteosomal degradation of Sp100		3.7	[29]
*4*	complement control protein CCPH		1.0	[33]
*11*	homolog to Raji LF2 of Epstein Barr virus		1.2	[32]
*12*	putative regulatory function		0.5	[32]
*13*	viral interleukin 17		0.5	[32]
*14*	viral superantigen		0.8	[31,32]
*15*	complement control protein, viral CD59		0.4	[53]
*16*	viral Bcl-2, apoptosis inhibition		0.5	[33]
*21*	thymidine kinase		0.9	
*45*	unknown	replication	0.8	
*49*	unknown	replication	0.9	
*52*	unknown	replication	0.3	
*62*	probable capsid assembly and maturation protein	(replication)	1.0	
*63*	tegument protein, NLR homolog	(replication)	2.7	
*64*	large tegument protein	replication	7.4	
*69*	unknown	replication	0.8	
*70*	thymidylate synthase	replication	0.9	
*71*	viral FLICE interacting protein, apoptosis inhibition		0.5	[30]
*72*	viral cyclin, inhibition of cell cycle arrest		0.8	[54]
*73*	latency-associated nuclear antigen homolog	persistence in proliferating cells	1.5	[54]
*74*	constitutively active G-protein coupled receptor		1.0	[52,55]
*75*	vFGARAT		3.9	[26]

## Data Availability

Not applicable.

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
