# Peer review of "Self-Repairing Herpesvirus Saimiri Deletion Variants"

_viruses, 2022, doi:10.3390/v14071525_

Round 1

Reviewer 1 Report

The authors have addressed my minor concerns

Author Response

No further alterations have been done, since specific issues have not been made. However, the text has been improved at a series of positions.

Reviewer 2 Report

Herpesvirus saimiri (HVS) has several advantages as a possible vector in gene therapy. This articles describes the production of a self-repairing vector by removing F plasmid derived sequences from the viral genome upon reconstitution in permissive epithelial cells. The system is capable of easily generating deletion variants to minimise the viral genome, with the longer term aim of creating a minimal virus vector for T cell gene therapy applications.

The manuscript is extremely well written and presented. Results are well controlled and consistent with findings described.

Results in Fig 2 and 3 are clear and genomic monitoring and replication properties are consistent between wild type viruses and virus variants. However, the differences in banding patterns in Fig 3 requires a little clearly explanation. In addition, please comment on the, if any, impact of the mutations in Table 1.

It is highly promising that human T cell can be transduced with the variants and continuously cultivated for 32 months. Moreover results shown in Fig 5 suggest limited if any effect on apoptosis induction by oxidative stress or topoisomerase inhibition on human T cells transformed with the variant viruses. These are exciting results and a short section in the discussion discussing possible future applications / diseases these vectors are working towards would enhance the manuscript.

Author Response

Thank you very much for the enthusiastic comments. The description of the band differences in Fig. 3 has been extended and the role of the sequence alterations has been described in more detail. The discussion includes now further aspects of immune gene therapy of SCID diseases.

This manuscript is a resubmission of an earlier submission. The following is a list of the peer review reports and author responses from that submission.

Round 1

Reviewer 1 Report

In this manuscript, Heyn et al set out to improve an HVS vector to possibly expand the gene therapy capacity of HVS. The authors generated a self-repairing minimal variants of the HVS C488. They chose  two viral genes to delete to design a HVS vector that may have less side effects and possibly less spread of the virus (ORF22 encoding the viral glycoprotein H crucial for entry into the host cell, and ORF29, which codes for the terminase). They show that their recombination protocol is effective, does not produce unexpected results, that the newly generated variants have viable kinetics in OMK cells. While this article appears scientifically sound, it is very method-oriented and only marginally improve upon prior studies. Furthermore, this article is written for an audience that is very much familiar with HVS biology as the authors do no explain thoroughly their reasoning behind targeting these particular viral genes, nor what L- and H-DNA is. 

Minor points: 

Fig 2b: the authors mention that the kinetics were performed on 3 biological replicates: they should thus show the error bars on the graph

The green/orange color scheme of figure one might not be the best choice for people with color blindness
